# Gut Microbiota and Metabolites May Play a Crucial Role in Sea Cucumber *Apostichopus japonicus* Aestivation

**DOI:** 10.3390/microorganisms11020416

**Published:** 2023-02-07

**Authors:** Yuan-Huan Kang, Bin-Tong Yang, Ren-Ge Hu, Peng Zhang, Min Gu, Wei Cong

**Affiliations:** Marine College, Shandong University, Weihai 264209, China

**Keywords:** *Apostichopus japonicus*, aestivation, gut microbiota, metabolites, 5-hydroxytryptophan

## Abstract

The constant increase in temperatures under global warming has led to a prolonged aestivation period for *Apostichopus japonicus*, resulting in considerable losses in production and economic benefits. However, the specific mechanism of aestivation has not been fully elucidated. In this study, we first tried to illustrate the biological mechanisms of aestivation from the perspective of the gut microbiota and metabolites. Significant differences were found in the gut microbiota of aestivating adult *A. japonicus* (AAJSD group) compared with nonaestivating adult *A. japonicus* (AAJRT group) and young *A. japonicus* (YAJRT and YAJSD groups) based on 16S rRNA gene high-throughput sequencing analysis. The abundances of Desulfobacterota, Myxococcota, Bdellovibrionota, and Firmicutes (4 phyla) in the AAJSD group significantly increased. Moreover, the levels of *Pseudoalteromonas*, *Fusibacter*, *Labilibacter*, *Litorilituus*, *Flammeovirga*, *Polaribacter*, *Ferrimonas*, *PB19,* and *Blfdi19* genera were significantly higher in the AAJSD group than in the other three groups. Further analysis of the LDA effect size showed that species with significant variation in abundance in the AAJSD group, including the phylum *Firmicutes* and the genera *Litorilituus*, *Fusibacter,* and *Abilibacter*, might be important biomarkers for aestivating adult *A. japonicus*. In addition, the results of metabolomics analysis showed that there were three distinct metabolic pathways, namely biosynthesis of secondary metabolites, tryptophan metabolism, and sesquiterpenoid and triterpenoid biosynthesis in the AAJSD group compared with the other three groups. Notably, 5-hydroxytryptophan was significantly upregulated in the AAJSD group in the tryptophan metabolism pathway. Moreover, the genera *Labilibacter*, *Litorilituus*, *Ferrimonas*, *Flammeovirga*, *Blfdi19*, *Fusibacter*, *Pseudoalteromonas,* and *PB19* with high abundance in the gut of aestivating adult *A. japonicus* were positively correlated with the metabolite 5-HTP. These findings suggest that there may be potential biological associations among the gut microbiota, metabolites, and aestivation in *A. japonicus*. This work may provide a new perspective for further understanding the aestivation mechanism of *A. japonicus.*

## 1. Introduction

*Apostichopus japonicus* (*A. japonicus*) has become one of the core species of sea cucumber in China and around the world [1]. With the increasing market demand for A. japonicus, the scale and breeding density of *A. japonicus* have constantly increased in China. Under the current global climate change, the sustainable development of the *A. japonicus* industry has been seriously affected. High-temperature stimulation leads to abnormal conditions, such as a decline in the growth rate, metabolic disorder, and prolongation of the aestivation period of *A. japonicus* [1,2]. During the aestivation period, *A. japonicus* stops feeding, its gut shrinks and degenerates, and its body weight significantly decreases, which prolongs the breeding cycle and greatly increases the breeding cost. Therefore, elucidating the aestivation mechanism of *A. japonicus* is of great significance for the breeding of superior varieties of *A. japonicus* and the sustainable development of *A. japonicus* aquaculture.

Research regarding the aestivation mechanism of *A. japonicus* is increasing. Researchers have studied the internal regulatory mechanism of aestivation not only from the perspectives of not only behavior, physiological and metabolic characteristics, and critical temperature but also from differentially expressed genes and proteins, miRNAs, genome methylation, and protein modification [3,4,5,6,7,8,9]. Among these studies, those including whole-genome sequencing and transcriptomic analysis of *A. japonicus* have provided important reference information for further revealing the aestivation mechanism of *A. japonicus* and promoting research progress [10]. Nonetheless, the aestivation of *A. japonicus* is considered to be a very complex biological process, and it is necessary to explore the internal regulatory mechanism of aestivation from different perspectives.

However, relevant studies on mammals may provide us with a reference, which found that gut microbes and metabolites were closely related to sleep or dormancy.

The gut microbiota and metabolites can directly or indirectly affect sleep through the “microbiota-gut-brain” axis [11]. There is significant evidence that the host gut microbiota not only affects digestive, metabolic, and immune functions but also regulates sleep and mental state through the “microbiota-gut-brain” axis [11]. Available evidence suggests that the gut microbiota interacts with circadian rhythm genes, and the characteristics of gut microbiota and metabolism are associated with host sleep and circadian rhythms [12]. Previous studies have found that the gut microbiota can affect sleep by regulating various factors, such as serotonin (a precursor of melatonin) synthesis or immune pathways [13]. Disruption of the gut microbiota leads to the elimination of serotonin from the gut and affects serotonin levels in the brain, which in turn affects the sleep/wake cycle [14]. However, during the aestivation period, the gut of *A. japonicus* degenerates and atrophies, and the metabolic rate decreases. This may result in the role of the gut and gut microbiota in aestivation being overlooked.

Thus, the following question arises: is there a biological relationship between the gut microbiota, metabolites, and aestivation in *A. japonicus*? To answer this question, we analyzed the gut microbiota of aestivating adults, nonaestivating adults, and young *A. japonicus* growing under different temperature conditions. At the same time, a comparative analysis of gut microbiota host cometabolites was carried out. This work may provide a new perspective for further understanding the aestivation mechanism of *A. japonicus*.

## 2. Materials and Methods

### 2.1. Experimental Animals, Grouping, and Feeding Conditions

The experimental *A. japonicus* were purchased from Weihai Xigang Aquatic Products Co., Ltd. The study included 24 young *A. japonicus* (4 months) with an average weight of 4.45 ± 1.14 g and 24 adult *A. japonicus* (24 months) with an average weight of 91.54 ± 3.63 g.

Before the experiment, the animals were acclimated for one month in a 600 L tank at 15 °C, during which they were fed mixed feed once a day. The feed mainly included 30% sea cucumber compound feed, 40% fresh sea mud, and 30% *Sargassum thunbergii* (Haida, China). In the nonaestivating group, named AAJRT, the growth temperature of adult *A. japonicus* was controlled at 15 °C. In the aestivation group, named AAJSD, the growth temperature of adult *A. japonicus* was slowly increased from 15 °C to 26 °C at a rate of 0.5 °C per day, inducing it to enter the state of aestivation. In addition, considering that young *A. japonicus* do not aestivate, we named the young *A. japonicus* growing at 15 °C the YAJRT group. In the YAJSD group, the growth temperature of young *A. japonicus* was slowly increased from 15°C to 26 °C at a rate of 0.5 °C. During the feeding period, the salinity was maintained at 29–32, the dissolved oxygen content was approximately 6–8 mg/L, and the pH was 8.0–8.5. The whole experimental period was 60 days.

### 2.2. Analysis of Weight Change

To clarify the effects of aestivation on the growth of adult *A. japonicus* and the effects of different growth temperatures on the growth of young *A. japonicus*, we recorded and analyzed the changes in the feed intake, movement status, and body weight of individuals in each group during the feeding process.

### 2.3. Microbial Diversity Analysis

The gut microbiotas of adult *A. japonicus* in the AAJSD group and AAJRT group were analyzed by 16S rRNA gene high-throughput sequencing. Similarly, the gut microbiotas of the young *A. japonicus* in the YAJRT group and YAJSD group were compared and analyzed under different growth temperatures.

The gut microbiotas of the AAJSD, AAJRT, YAJRT, and YAJSD groups were examined and comparatively analyzed by high-throughput 16S rRNA gene sequencing. Twelve *A. japonicus* were randomly selected from each group after seven days of feeding under different feeding conditions. The gut contents were collected, flash-frozen in liquid nitrogen, and then stored at −80 °C; half of these samples were used for microbial diversity analysis and the other half for metabolomic analysis. The detailed steps were as follows.

#### 2.3.1. Sequencing

The cetyltrimethylammonium bromide (CTAB) method was used to extract the total genomic DNA from six samples. One-percent agarose gels were used to evaluate DNA concentrations and purity. Using sterile water, the DNA was diluted to 1 ng/µL. The V3-V4 regions of the 16S rRNA genes were amplified using specific primers (338F-806R). PCRs and purification of PCR products were performed with reference to the literature [15]. Following the manufacturer’s recommendations, sequencing libraries were generated with the NEB Next^®^ Ultra™ IIDNA Library Prep Kit (New England Biolabs, USA). The library quality was evaluated on a Qubit@ 2.0 Fluorometer (Thermo Scientific, USA) and Agilent Bioanalyzer 2100 system (Agilent Technologies, USA). Finally, the library was sequenced on an Illumina NovaSeq platform, and 250 bp paired-end reads were generated [16,17].

#### 2.3.2. Data Analysis

Paired-end reads were assigned to samples based on their unique barcodes and were truncated by cutting off the barcodes and primer sequences. Paired-end reads were merged using FLASH (version 1.2.11, http://ccb.jhu.edu/software/FLASH/ (accessed on 20 December 2021)) [18], and the splicing sequences were called raw tags. The filtering of the raw tags and comparative analysis of the clean tags were carried out with specific references to the literature [19].

For the obtained effective tags, denoising was performed with the DADA2 module in QIIME2 software (Version QIIME2-202006) to obtain initial amplicon sequence variants (ASVs), and then ASVs with abundances less than 5 were filtered out [20]. Species annotation was performed using QIIME2 software with the Silva Database. Phylogenetic relationship construction and data normalization were performed according to previously reported methods [21].

To analyze the diversity, richness, and uniformity of the communities in the samples, alpha diversity was calculated from indices in QIIME2, including the ASV, Chao1, Shannon, Simpson, Dominance, and Good coverage indices. To evaluate the complexity of the community composition and compare the differences between groups, beta diversity was calculated based on weighted and unweighted UniFrac distances in QIIME2. Cluster analysis was performed with principal component analysis (PCA), which was applied to reduce the dimensions of the original variables using the ade4 package and ggplot2 package in R software (Version 3.5.3). Principal coordinate analysis (PCoA) was performed to obtain principal coordinates and visualize differences in samples in complex multidimensional data. The two-dimensional PCoA results were displayed using the ade4 package and ggplot2 package in R software (Version 2.15.3). QIIME2 and LEfSe software were used to study the significance of the differences in the community structure of each group and the biomarkers. PICRUSt2 software was used for the annotation of community functions. The specific methods were performed with reference to the literature [22].

### 2.4. Metabolomic Analysis

#### 2.4.1. Metabolite Extraction

To extract metabolites, the samples (six samples per group) were placed in EP tubes and resuspended in prechilled (4 °C) 80% methanol by vortexing. Then, the samples were melted on ice and centrifuged for 30 s. After sonication for 6 min, the samples were centrifuged at 5000 rpm and 4 °C for 1 min. The supernatant was freeze-dried and dissolved in 10% methanol. Finally, the solution was injected into the LC-MS/MS system for analysis [23,24].

#### 2.4.2. UHPLC-MS/MS Analysis

UHPLC-MS/MS analyses were performed using a Vanquish UHPLC system (ThermoFisher, Germany) coupled with an Orbitrap Q ExactiveTM HF mass spectrometer (ThermoFisher, Germany) by Novogene Co., Ltd. (Beijing, China). Samples were injected onto a Hypesil Gold column (100 × 2.1 mm, 1.9 μm) using a 17 min linear gradient at a flow rate of 0.2 mL/min. The eluents for the positive polarity mode were eluent A (0.1% FA in water) and eluent B (methanol). The eluents for the negative polarity mode were eluent A (5 mM ammonium acetate, pH 9.0) and eluent B (methanol). The Q ExactiveTM HF mass spectrometer was operated in positive/negative polarity mode with a spray voltage of 3.5 kV, a capillary temperature of 320 °C, a sheath gas flow rate of 35 psi, an auxiliary gas flow rate of 10 L/min, an S-lens RF level of 60, and an auxiliary gas heater temperature of 350 °C.

#### 2.4.3. Data Processing, Metabolite Identification, and Data Analysis

The raw data files generated by UHPLC-MS/MS were processed using compound Discoverer 3.1 (ThermoFisher, Germany) to perform peak alignment, peak picking, and quantitation for each metabolite. After that, peak intensities were normalized to the total spectral intensity. The normalized data were used to predict the molecular formula based on additive ions, molecular ion peaks, and fragment ions. Then, peaks were matched with the mzCloud, mzVault, and MassList databases to obtain accurate qualitative and relative quantitative results. Statistical analyses were performed using the statistical software R (version R-3.4.3), Python (version 2.7.6), and CentOS (version 6.6). When data were not normally distributed, normal transformations were attempted using the area normalization method.

These metabolites were annotated using the KEGG database (https://www.genome.jp/kegg/pathway.html (accessed on 25 December 2021)). Principal component analysis (PCA) and partial least squares discriminant analysis (PLS-DA) were performed with metaX [25]. *P*-values were calculated using univariate analysis (*t*-test). Differential metabolites had VIP > 1, *p*-value < 0.05, and fold change (FC) ≥2 or FC ≤ 0.5. Volcano plots and heatmaps were created by a hierarchical clustering method based on the literature [25]. Statistically significant correlations between differential metabolites were calculated in R language. A *p*-value < 0.05 was considered to indicate statistical significance, and correlation plots were prepared by the corrplot package in R language. The functions of these metabolites and metabolic pathways were studied using the KEGG database. Metabolic pathway enrichment of differential metabolites was performed when the ratio satisfied the criterion x/n > y/N (where N is the number of identified metabolites that matched KEGG pathways; n is the number of differential metabolites in N; y is the number of metabolites that were annotated to a certain KEGG pathway; and x is the number of differential metabolites that were annotated to a certain KEGG pathway). Metabolic pathways were considered significantly enriched when their *p*-value was <0.05.

### 2.5. Correlation Analysis

To determine the association between 16S rRNA high-throughput sequencing results and metabolomics results, correlation analysis was performed based on the Pearson correlation coefficient between the top 16 significantly differentially abundant genera and the top 20 significant differential metabolites. In particular, the relationship between the differentially abundant microbes and metabolites in the gut of aestivating and nonaestivating adult *A. japonicus* (AAJSD vs. AAJRT group) was analyzed. The Spearman coefficient correlation (rho) and *p*-value were calculated for the relative abundance of each differentially abundant genus and between the different differential metabolites, and a *p*-value  <  0.05 (*) was considered to indicate a significant difference.

## 3. Results

### 3.1. The Effect of High Temperature on the Growth of Adult and Young A. Japonicus

Changes in body weight were compared before and after aestivation in adult *A. japonicus*. The analysis results showed that after aestivation, the body weight of adult *A. japonicus* (AAJSD group) decreased from 91.54 ± 3.63 g before aestivation to 52.32 ± 5.29 g after aestivation, a decrease of approximately 42.84%, and the weight loss was very obvious (as shown in Figure 1A). After 30 days of feeding, the body weight of nonaestivating adult *A. japonicus* (AAJRT group) increased from 81.20 ± 2.39 g to 87.61 ± 4.72 g, with normal growth and a significant increase in body weight (as shown in Figure 1A). The results showed that aestivation could lead to significant weight loss in adult *A. japonicus*.

In addition, the effects of different growth temperatures on the growth of young *A. japonicus* were compared and analyzed. At a growth temperature of 26 °C, the young *A. japonicus* (YAJSD group) exhibited normal feeding behavior but did not enter the state of aestivation, and the body weight increased from 4.04 ± 0.88 g to 7.20 ± 3.25 g (Figure 1B). The body weight of young *A. japonicus* in the YAJRT group (growth temperature was 15 °C) increased from 4.45 ± 1.14 g to 11.55 ± 3.17 g (Figure 1B), and the weight growth rate was higher than that in the YAJSD group. The results showed that the increase in temperature could not induce the young *A. japonicus* to enter the state of aestivation, and the high temperature had an important effect on the growth of the young *A. japonicus*. Compared with 26 °C, 15 °C was more suitable for the growth of young *A. japonicus*.

### 3.2. Microbial Diversity Analysis

High-throughput sequencing analysis of the 16S rRNA gene was performed to analyze the gut microbiota of adult *A. japonicus* in the AAJSD and AAJRT groups. The results showed that there were significant differences in the gut microbiota between aestivating and nonaestivating adult *A. japonicus*. In addition, a comparative analysis of the gut microbiota of young *A. japonicus* under different growth temperature conditions showed that there were also differences between the young *A. japonicus* of the YAJRT group and those of the YAJSD group. Considering that young *A. japonicus* do not aestivate, we focused on the comparison and analysis of the gut microbiota differences between the AAJSD group and the other three groups (AAJRT, YAJRT, and YAJSD).

Based on the ASVs obtained after noise reduction, the common and unique ASVs among different groups were analyzed. A total of 335 ASVs were shared by the four experimental groups; 513 ASVs were unique to the YAJRT group, 729 ASVs were unique to the YAJSD group, 294 ASVs were unique to the AAJRT group, and 412 ASVs were unique to the AAJSD group (Figure 2A). To further study the phylogenetic relationships of species at the genus level, the representative sequences of the top 100 genera for each group were obtained by multiple sequence alignment, and the constructed phylogenetic tree is shown in Figure 2B. The AAJRT, YAJRT, and YAJSD groups were used as controls, and we focused on the comparative analysis of the differences in the gut microbiota of aestivating adult *A. japonicus* (AAJSD group) and controls. The analysis results showed that at the phylum level, the abundances of Campylobacterota, Cyanobacteria, Patescibacteria, etc., were significantly decreased in the gut of aestivating adult *A. japonicus*, while the abundances of Firmicutes, Desulfobacterota, Myxococcota, etc., were significantly increased (Figure 2C). At the genus level, the abundances of *Halarcobacter*, *Cobetia*, *Pseudomonas*, *Lutibacter*, *Haloferula*, *Lutimonas, Actibacter*, etc., in the gut of aestivating adult *A. japonicus* were significantly decreased, while the abundances of *Fusibacter*, *Labilibacter*, *Litorilituus*, etc., were significantly increased (Figure 2D).

Similarly, the results of further species abundance cluster analysis showed that the abundances of Desulfobacterota, Myxococcota, Bdellovibrionota, and Firmicutes in the gut of aestivating adult *A. japonicus* (AAJSD group) were significantly higher than those in the other three groups (Figure 2E). At the genus level, the abundances of *Pseudoalteromonas, Fusibacter, Labilibacter, Litorilituus, Flammeovirga, Polaribacter, Ferrimonas, PB19,* and *Blfdi19* in the AAJSD group were significantly higher than those in the other three groups (Figure 2F).

The alpha diversity analysis suggested that there were significant differences in the richness indices (Shannon and Simpson) between the YAJ (RT and SD) and AAJ (RT and SD) groups. The richness indices of the gut microbiota in young *A. japonicus* (YAJRT and YAJSD groups) were higher than those in adult *A. japonicus* (AAJRT and AAJSD groups) (Figure 3A,B). However, there were significant differences within the YAJ group, and the same was true for the AAJ group. Notably, the richness of the gut microbiota of young *A. japonicus* (YAJSD group) and adult *A. japonicus* (AAJSD group) increased with increasing growth temperature. This was proven by beta diversity analysis performed using PCoA and nonmetric multidimensional scaling (NMDS) (Figure 3C,D).

LDA effect size (LEfSe) was used to analyze the species with significant differences in abundance among the four experimental groups to find biomarkers with significant differences. As seen from the bar chart of the LDA value distribution and the evolutionary branching chart (Figure 3E,F), there were species with significant variation in each group. We focused on species with significant variation in abundance in the AAJSD group, including Firmicutes at the phylum level and *Litorilituus*, *Fusibacter,* and *Abilibacter* at the genus level. These species may be important biomarkers for aestivating adult *A. japonicus*.

To further understand the functions of species with significant variation between groups, we performed a functional prediction analysis using the KEGG orthology (KO) database. Based on the functional annotations and abundance information of the samples in the KO database, the top 35 functions in terms of abundance and their abundance information in each group were selected for heatmap and clustering (Appendix A). Dimensionality reduction was performed using PCA (Appendix A). The results of the analysis showed that there were significant differences in the abundance of KO functional categories among the groups. We focused on the KO functional categories with relatively high abundance (K02003, K02004, K01897, K02014, and K01992) (Appendix A) and 20 unique KO functional categories (Appendix A) in the AAJSD group. These KO functional categories were mainly related to sugar metabolism, fat metabolism, and energy metabolism. This may be because *A. japonicus* cannot obtain external energy and can rely only on its own metabolism to maintain energy needs during aestivation. This also seems to explain why the weight of *A. japonicus* dropped significantly during aestivation.

### 3.3. Metabolomics Analysis

In addition to the high-throughput sequencing analysis of the 16S rRNA gene, we performed untargeted metabolomics analysis to reveal more metabolomics features and differences of *A. japonicus* between the AAJSD group and the other three groups. There were significant differences in gut microbiota host cometabolites between aestivating (AAJSD group) and nonaestivating (AAJRT group) adult *A. japonicus*, and a total of 339 differential metabolites were identified. Compared with the AAJRT group, there were 228 upregulated metabolites and 111 downregulated metabolites in the AAJSD group (Figure 4A). Similarly, a comparative analysis of the gut microbiota host cometabolites of young *A. japonicus* under different growth temperature conditions showed that there were also differences between the YAJRT group and the YAJSD group, and a total of 216 differential metabolites were identified. Compared with the YAJRT group, 114 metabolites were upregulated and 102 were downregulated in the YAJSD group (Figure 4B). In addition, we further compared and analyzed the differences in gut microbiota host cometabolites between the AAJSD group and the YAJSD group. The results showed that there were also significant differences between the AAJSD group and the YAJSD group, and a total of 302 differential metabolites were identified. Compared with the YAJSD group, there were 185 upregulated and 117 downregulated differential metabolites in the AAJSD group (Figure 4C). Moreover, there were 369 differential metabolites between the AAJSD group and the YAJRT group. In the AAJSD group, there were 243 upregulated and 126 downregulated differential metabolites compared with the YAJRT group (Figure 4D).

Hierarchical clustering analysis (HCA) of differential metabolites was carried out, and there were obvious differences among all the groups (Figure 4E,F). The results marked with the names of the differential metabolites and the raw data are shown in Appendix A. The KEGG pathway enrichment analysis of the above differential metabolites showed that there were three differential metabolic pathways in the AAJSD group, namely biosynthesis of secondary metabolites, tryptophan metabolism, and sesquiterpenoid and triterpenoid biosynthesis, compared with the other three groups (Figure 5A–C). However, these differences were not found between the YAJRT and YAJSD groups (Figure 5D). Among them, in the biosynthesis of secondary metabolites pathway, the upregulated metabolites in the AAJSD group were as follows: porphobilinogen, gluconic acid lactone, xanthosine, riboflavin, lactoflavin, 7,8-dimethyl-10-ribitylisoalloxazine, vitamin B2, phenylpyruvic acid, alpha-ketohydrocinnamic acid, keto-phenylpyruvate, 3-phenyl-2-oxopropanoate, 2-Oxo-3-phenylpropanoate, L-phenylalanine, and (S)-alpha-amino-beta-phenylpropionic acid. Notably, in the tryptophan metabolism pathway, 5-hydroxytryptophan (5-HTP) was significantly upregulated in the AAJSD group (Table 1). The results indicated that tryptophan metabolism was significantly enhanced in the gut of aestivating adult *A. japonicus* (AAJSD group).

### 3.4. Correlation between Differentially Abundant Microbes and Differential Metabolites

Correlation analysis was performed based on the Pearson correlation coefficient between the top 16 significantly differentially abundant genera and the top 20 significantly differentially abundant metabolites in the AAJSD group compared with the AAJRT group. There were significant correlations between the differentially abundant microbes and differential metabolites (Figure 6A). Among them, the metabolite prostaglandin E2-1-glyceryl ester had no significant correlation with the top 16 genera. The metabolite 1-(2,4-dihydroxyphenyl)-2-(3,5-dimethoxyphenyl) propan-1-one was found to be associated only with *Erythrobacter*, *Agarivorans*, *Pseudoalteromonas,* and *PB19*. The genera *Endozoicomonas*, *Halarcobacter,* and *Pseudophaeobacter* were negatively correlated with all top 20 differential metabolites. However, the other 12 genera were positively correlated with the metabolites. Notably, eight genera (*Labilibacter*, *Litorilituus*, *Ferrimonas*, *Flammeovirga*, *Blfdi19*, *Fusibacter*, *Pseudoalteromonas,* and *PB19*) with high abundance were positively correlated with the metabolite 5-HTP in the gut of aestivating adult *A. japonicus* (AAJSD group). Moreover, these genera are closely related to each other, and they may play important biological roles in the gut of aestivating adult *A. japonicus* (Figure 6B).

## 4. Discussion

Previous studies have shown that high temperature is the main cause of aestivation in *A. japonicus* [26]. However, there has been no breakthrough in how high temperature triggers this regulatory mechanism. The aestivation of *A. japonicus* is considered to be a very complex biological process. It is necessary to explore the internal regulatory mechanism of aestivation from different perspectives.

In this study, we explored the aestivation mechanism of *A. japonicus* from the perspective of the gut microbiota and metabolites. The richness of the gut microbiota tended to increase with increasing growth temperature in both young *A. japonicus* and adult *A. japonicus*. Moreover, the richness index of the gut microbiota in young *A. japonicus* was significantly higher than that in adult *A. japonicus*. In addition, we found that there were significant differences in the gut microbiota and metabolites of aestivating adult *A. japonicus* compared with nonaestivating adult *A. japonicus* and young *A. japonicus*, indicating that there were obvious biological correlations among the gut microbiota, metabolites, and aestivation for *A. japonicus*. The analysis results showed that at the phylum level, the abundances of Campylobacterota, Cyanobacteria, and Patescibacteria were significantly decreased in the gut of aestivating adult *A. japonicus*, while the abundances of Firmicutes, Desulfobacterota, Myxococcota, and Bacteroidetes (also known as Bacteroidota) were significantly increased. A study on the composition of the human intestinal microbiome showed that the abundance of Bacteroidetes and Firmicutes was positively correlated with the quality of human sleep [27]. This finding is similar to the results of this study. We also obtained similar results at the genus level: the abundances of *Halarcobacter*, *Cobetia*, *Pseudomonas*, *Lutibacter*, *Haloferula*, *Lutimonas,* and *Actibacter* in the gut of aestivating adult *A. japonicus* were significantly decreased, while the abundances of *Pseudoalteromonas*, *Fusibacter*, *Labilibacter*, *Litorilituus*, *Flammeovirga*, *Polaribacter*, *Ferrimonas*, *PB19,* and *Blfdi19* were significantly increased. This indicates that Bacteroidetes (e.g, *Flammeovirga*, *Labilibacter,* and *Polaribacter*) and Firmicutes (e.g., *Fusibacter*) may play an important role in the aestivation of *A. japonicus*. Additionally, considering that *A. japonicus* belongs to the echinoderm, there may be differences between its gut microbiota and that of mammals. Whether the other genera with high relative abundances (e.g., *Pseudoalteromonas*, *Litorilituus*, *Ferrimonas*, *PB19,* and *Blfdi19*) have the same function in the gut of *A. japonicus* is not known. The relationship between these significant changes in the gut microbiota and the aestivation of *A. japonicus* is still unclear and deserves further investigation. Thus, as mentioned above, these high-abundance genera are likely to be important biomarkers for aestivating adult *A. japonicus*.

In addition, analysis of KEGG metabolic pathways associated with the differential metabolites showed that secondary metabolite biosynthesis, tryptophan metabolism, and sesquiterpenoid and triterpenoid biosynthesis pathways in the gut of aestivating *A. japonicus* showed significant changes. Whether these changes are related to the aestivation of *A. japonicus* also needs to be further explored. Tryptophan metabolism was significantly enhanced in the gut of aestivating *A. japonicus*, and the expression of the tryptophan metabolite 5-HTP was significantly increased, which aroused our attention. Considering the characteristics of young *A. japonicus,* which do not aestivate, further comparative analysis showed that the tryptophan metabolism level in the gut of young *A. japonicus* under different growth temperatures did not change significantly, and this change was observed only in the gut of aestivating *A. japonicus*. This suggests that there were significant differences in tryptophan metabolism between young *A. japonicus* and adult *A. japonicus*. This indicated that tryptophan metabolism might be closely related to the aestivation of *A. japonicus*. The research on the mechanism of tryptophan metabolism in *A. japonicus* is very limited. Currently, the mechanism of tryptophan metabolism in *A. japonicus* is not clear, and whether there are differences in the tryptophan metabolic pathway between echinoderms and mammals and whether the function of its metabolite 5-HTP is the same are also unknown. In addition, the biological function of 5-HTP in the aestivation of *A. japonicus* has not been elucidated.

However, relevant studies on mammals may provide us with a reference, and it was found that 5-HTP supplementation could significantly improve human sleep quality [28]. Similarly, another study found that inhibition of 5-HTP or 5-hydroxy tryptamine (5-HT) synthesis by tryptophan hydroxylase inhibitors significantly affected sleep in rats. In contrast, adding tryptophan or 5-HTP could significantly improve the sleep quality of rats [29]. These studies indicate that the tryptophan metabolite 5-HTP/5-HT extensively affects the sleep and dormancy of animals. Similar findings have been found in studies on the dormancy mechanism in other animals. A previous study indicated that tryptophan metabolism in red blood cells was significantly enhanced, the total tryptophan and free tryptophan levels in plasma were significantly decreased, and the concentration of related metabolites was significantly increased during the hibernation of arctic ground squirrels [30]. However, the expression of a tryptophan hydroxylase activator (tryptophan 5-monooxygenase activated protein) in the brains of *African lungfish* is significantly upregulated during aestivation, which may lead to an increase in the level of the downstream metabolite 5-HT, which in turn participates in the regulation of the sleep–wake cycle [31]. In addition, a study on *Citellus erythrogenys* showed that the activity of tryptophan hydroxylase (the key rate-limiting enzyme for the conversion of tryptophan to 5-HTP) was activated in the brain tissue before the animal entered hibernation. The activity of tryptophan hydroxylase increased significantly after hibernation and decreased significantly after awakening. These results suggest that 5-HTP/5-HT may be closely related to the transformation and maintenance of hibernation in *C. erythrogenys* [32]. These studies indicate that the tryptophan metabolites 5-HTP and 5-HT extensively affect the sleep and dormancy of animals.

Moreover, studies on tryptophan metabolism in mammals have found that tryptophan metabolism is closely related to the gut microbiota. There are three main metabolic pathways for tryptophan metabolism in the mammalian gut: (1) the gut microbiota directly converts tryptophan to indole and its derivatives [33,34,35,36]; (2) tryptophan is metabolized in gut immune cells and epithelial cells to form kenurine [37]; and (3) tryptophan is metabolized in gut enterochromaffin cells to produce 5-HTP/5-HT [38]. In addition to directly participating in the first metabolic pathway, the gut microbiota is also a major player in the metabolic production of 5-HTP/5-HT by gut enterochromaffin cells. Although the mechanism by which the gut microbiota regulates 5-HTP/5-HT production is not fully understood, this effect has been confirmed in germ-free mice. A study published in *Cell* found that short-chain fatty acids produced by spore-producing bacteria in the gut of humans and mice could stimulate the expression of the tryptophan hydroxylase TpH1 in gut enterochromaffin cells and promote the synthesis of 5-HTP and 5-HT. This indicated that the intestinal microbiota was involved in the regulation of 5-HTP/5-HT biosynthesis [38]. Moreover, previous studies have shown that *Candida*, *Streptococcus*, *Escherichia coli,* and *Enterococcus* can also directly participate in the metabolism of tryptophan and convert it to 5-HT [39]. In addition, 5-HTP in the gut can be transported to the central nervous system through the circulatory system, where it is further metabolized and involved in the regulation of many biological processes, including sleep, mood, and immunity [40]. Conversely, the host can also influence the composition and structure of the gut microbiota through the “microbiota-gut-brain axis”, and they can conduct two-way information communication and influence each other [41,42,43]. These studies indicate that the gut microbiota is widely involved in tryptophan metabolism and that there is an interaction between the two factors.

To determine whether these differentially abundant microbes in the gut of aestivating *A. japonicus* were related to tryptophan metabolism, we further conducted a correlation analysis between 16S rRNA gene high-throughput sequencing and metabolomics results and found that there was a significant correlation between the differentially abundant microbes and tryptophan metabolism. Notably, eight bacterial genera (*Labilibacter*, *Litorilituus*, *Ferrimonas*, *Flammeovirga*, *Blfdi19*, *Fusibacter*, *Pseudoalteromonas,* and *PB19*) with high abundance in the gut of aestivating adult *A. japonicus* were positively correlated with the metabolite 5-HTP. Previous studies have shown the probable effects of the five phyla Actinobacteria, Firmicutes, Proteobacteria, Bacteroidetes, and Fusobacteria on the tryptophan metabolism pathways in the human gut [44]. In this study, we found that *Fusibacter,* a member of the phylum Firmicutes, was significantly associated with tryptophan metabolism. This suggests that *Fusibacter* may play an important role in the metabolism of tryptophan in the gut of *A. japonicus*. However, *Labilibacter*, *Litorilituus*, *Ferrimonas*, *Flammeovirga*, *Blfdi19*, *Pseudoalteromonas,* and *PB19* genera may also be closely related to 5-HTP and aestivation in *Apostichopus japonicus*, although this has not been confirmed by other studies. Their functions deserve further in-depth study. These results suggest that the gut microbiota may be involved in the metabolism of tryptophan. Therefore, we speculate that the gut microbiota of *A. japonicus* may participate in the regulation of aestivation through the tryptophan metabolism pathway. Nevertheless, the limitation of this study is that the proposed inference has not been tested by relevant experiments. This, of course, will be the subject of subsequent research.

## 5. Conclusions

In conclusion, our results demonstrate for the first time that there is a significant biological association among the gut microbiota, metabolites, and aestivation in *A. japonicus*. Significant changes were found in the gut microbiota and metabolites of adult *A. japonicus* during aestivation. The genera *Fusibacter*, *Litorilituus*, *Abilibacter*, etc., might be important biomarkers for aestivating adult *A. japonicus*. Importantly, 5-hydroxytryptophan may play an important role in the aestivation of *A. japonicus*. These findings provide a new perspective and idea for further understanding the aestivation mechanism of *A. japonicus*.

## Figures and Tables

**Figure 1 microorganisms-11-00416-f001:**
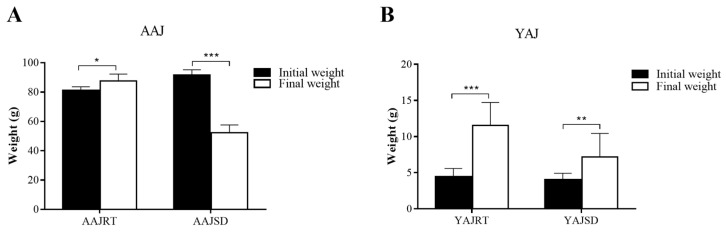
The effect of temperature on the growth of *A. japonicus.* (**A**) Effects of aestivation on the growth of adult *A. japonicus* (AAJ): AAJRT group, nonaestivating group (growth temperature 15 °C); AAJSD group, aestivation group (growth temperature 26 °C). (**B**) Effects of different temperatures on the growth of young *A. japonicus* (YAJ): YAJRT group, growth temperature 15 °C; YAJSD group, growth temperature 26 °C. *: *p*-value < 0.05; **: *p*-value < 0.01; ***: *p*-value < 0.001.

**Figure 2 microorganisms-11-00416-f002:**
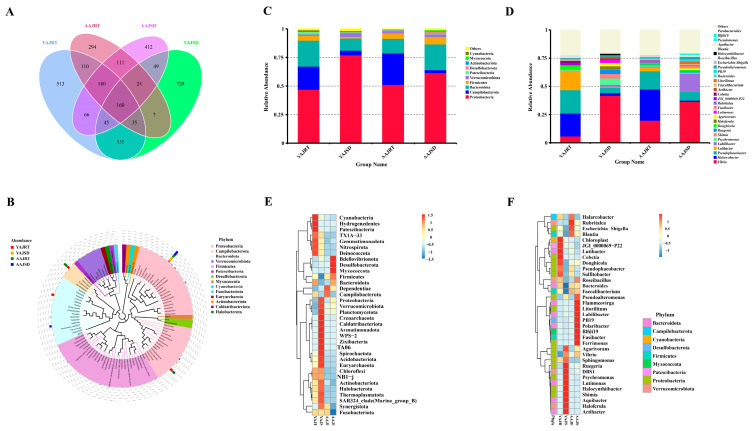
Analysis of the amplicon sequence variants (ASVs) in the gut microbiota of *A. japonicus* from different groups. (**A**) Quantitative analysis of shared and specific ASVs of the gut microbiota among different groups. (**B**) Phylogenetic relationships of species at the genus level; the representative sequences of the top 100 genera were obtained by multiple sequence alignment, and a phylogenetic tree of the representative sequences of the species at the genus level was constructed. The colors of the sectors and branches represent their corresponding phyla, and the stacked bar chart outside the sector ring represents the abundance distribution information of the genus in different samples. (The legend on the left is the sample information and the legend on the right is the classification information at the phylum level corresponding to the species at the genus level.) (**C**) Top 10 relative abundances of gut microbiota members in each group at the phylum level. (**D**) Top 30 relative abundances of gut microbiota members in each group at the genus level. (**E**) Cluster analysis of the species abundance of the gut microbiota at the phylum level. (**F**) Cluster analysis of the species abundance of the gut microbiota at the genus level.

**Figure 3 microorganisms-11-00416-f003:**
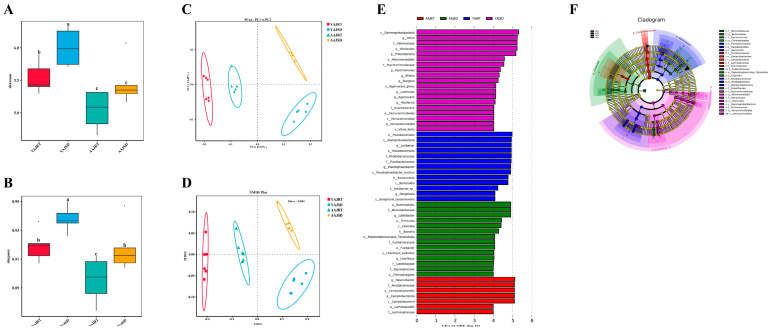
Analysis of microbial community diversity and differences in the abundance of species among different groups. Analysis of microbial community diversity. (**A**) Alpha diversity analysis of the gut microbiota (Shannon index). (**B**) Alpha diversity analysis of the gut microbiota (Simpson index). (**C**) Beta diversity analysis of the gut microbiota (PCoA, principal coordinates analysis using weighted UniFrac distance). (**D**) Beta diversity analysis of the gut microbiota (NMDS, nonmetric multidimensional scaling). Analysis of differences in species abundance by LefSe. (**E**) Histogram of LDA value distribution: the length of the bar graph represents the impact of different species (LDA score). Biomarkers with an LDA score greater than the set value (the default value was 4) are shown. (**F**) Evolutionary branching diagram: the circles radiating from inside to outside represent taxonomic levels from phylum to genus. Each small circle at a different taxonomic level represents a classification at that level, and the size of the small circle diameter is proportional to the relative abundance size. The species with no significant difference are uniformly colored yellow, and the different biomarker species followed the group to color. The red nodes represent microbial groups that play an important role in the red group, and the green nodes represent microbial groups that play an important role in the green group. If a group is missing in the figure, it indicates that there were no significant differentially abundant species in the group. Species names indicated by English letters are shown in the legend on the right. Different letters in Figure 3A,B indicate that the differences between the groups were statistically significant (Tukey’s test, *p*-value < 0.05).

**Figure 4 microorganisms-11-00416-f004:**
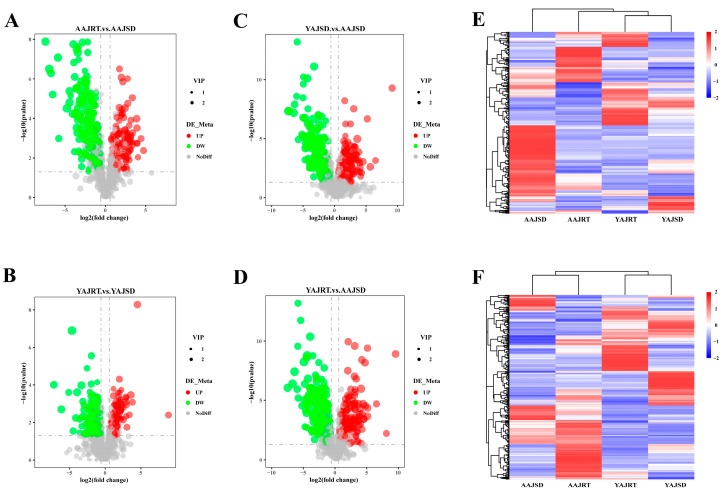
Difference analysis of the gut microbiota host cometabolites among different groups. (**A**) Differences in gut microbiota host cometabolites between the AAJRT and AAJSD groups. (**B**) Differences in gut microbiota host cometabolites between the YAJRT and YAJSD groups. (**C**) Differences in gut microbiota host cometabolites between the YAJSD and AAJSD groups. (**D**) Differences in gut microbiota host cometabolites between the YAJRT and AAJSD groups. (**E,F**) Clustering heatmap of the differential metabolites, negative ion mode (**E**), and positive ion mode. (**F**) Longitudinal is the clustering of samples, horizontal is the clustering of metabolites, and shorter clustering branches represent higher similarity.

**Figure 5 microorganisms-11-00416-f005:**
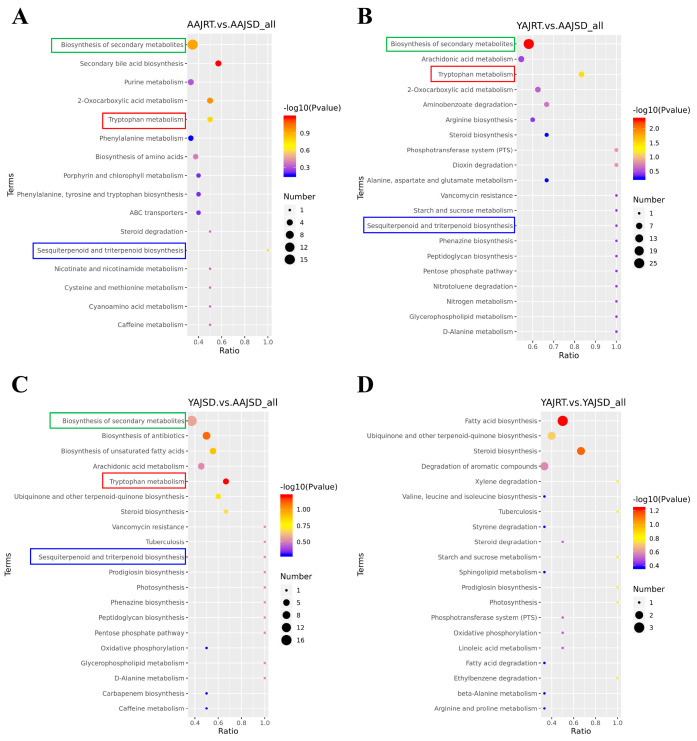
KEGG analysis of differential metabolites among different groups (top 20). (**A**) Results of KEGG analysis of differential metabolites between the AAJSD and AAJRT groups. (**B**) Results of KEGG analysis of differential metabolites between the AAJSD and YAJRT groups. (**C**) Results of KEGG analysis of differential metabolites between the AAJSD and YAJSD groups. (**D**) Results of KEGG analysis of differential metabolites between the YAJRT and YAJSD groups.

**Figure 6 microorganisms-11-00416-f006:**
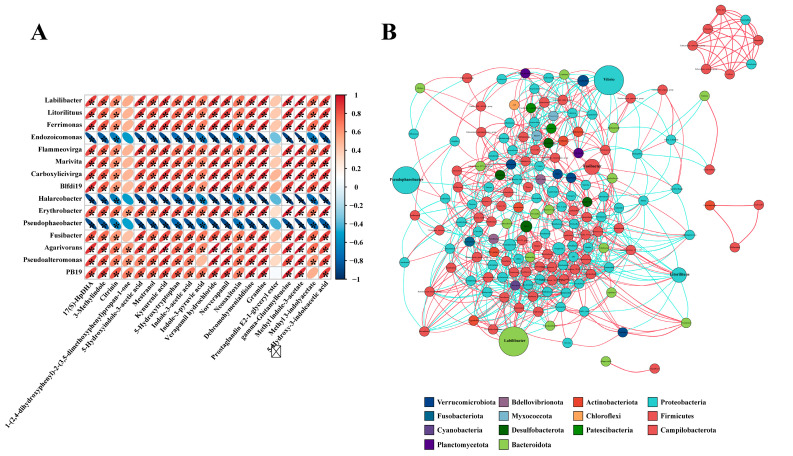
Correlation analysis between differentially abundant microbes and differential metabolites. (**A**) Correlation analysis was conducted based on the Pearson correlation coefficient between the significant differentially abundant genera obtained by high-throughput sequencing analysis of the 16S rRNA gene and the significant differential metabolites obtained by metabolomics analysis (AAJRT vs. AAJSD group). In the figure, the horizontal rows represent the differentially abundant genera, the vertical columns represent the differential metabolites, and the legend on the right is the correlation coefficient. Red indicates a positive correlation, blue indicates a negative correlation, and asterisk (*) marks indicate statistical significance, a *p*-value < 0.05. (**B**) Analysis of the association network of the dominant gut microbes in the AAJSD group. Different nodes represent different genera, node size represents the average relative abundance of the genus, nodes of the same clade have the same color (as shown in the legend), the thickness of the line between nodes is positively correlated with the absolute value of the correlation coefficient of species interactions, red lines indicate a positive correlation, and blue lines indicate a negative correlation.

**Table 1 microorganisms-11-00416-t001:** Comparative analysis of 5-hydroxytryptophan among different groups.

Group	Name	FC	*p*-Value	VIP	Up or Down
AAJSD vs. AAJRT	5-hydroxytryptophan	3.731	0.000309	1.617	Up
AAJSD vs. YAJSD	5-hydroxytryptophan	2.695	0.009288	1.103	Up
AAJSD vs. YAJRT	5-hydroxytryptophan	2.968	0.000472	1.504	Up

Note: The screening of differential metabolites was performed based mainly on the VIP, FC, and *p*-value. VIP refers to the variable importance in the projection of the first principal component of the PLS-DA model, which represents the contribution of metabolites to grouping; FC refers to the multiple of difference; *p*-value indicates the significance level of differences.

## Data Availability

All main data are provided in the main text or Appendix A.

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
