# Peer review of "Gut Microbiota and Metabolites May Play a Crucial Role in Sea Cucumber Apostichopus japonicus Aestivation"

_microorganisms, 2023, doi:10.3390/microorganisms11020416_

Round 1
Reviewer 1 Report
I suggest it's acceptanct in the reccent form.
Reviewer 2 Report
The stimulation of continued high temperatures has led to a prolonged aestivation period for Apostichopus japonicus, resulting in huge losses in production and economic benefits. Thus , the study of the specific mechanism of aestivation holds great significance.
This paper referred the relevant study on mammals, and got the similar results to that of mammals about aestivation. Overall, this paper is a high merit study.
In addition, there were several points needed to be verified.
1. The abbreviation of groups’ name is a little complicated, I suggest to simplify them;
2. Did the sea cucumber test their pathogens before conducting the design experiments?
3. Line 476~482, the authors highlighted the relevance of the 5-HTP for both humans and rats, yet the reference cites was too comprehensive to explain its mechanism. In this section, I suggest to add the explanation, then it would more interesting.
4. In discussion section, the discuss about the bacteria is relatively short. As for readers, the key bacteria related to 5-HTP and aestivation is more attractive. It is recommended that more discussion should be held on significant bacteria.
Reviewer 3 Report
Review for the paper "Gut microbiota and metabolites may play a crucial role in sea cucumber Apostichopus japonicus aestivation" by Yuan-Huan Kang, Bin-Tong Yang, Ren-Ge Hu, Peng Zhang, Min Gu, Wei Cong submitted to "Microorganisms".
General comment.
Aestivation is a special type of dormancy that allows animals to survive during periods of extreme heat or drought. Studies on aestivation physiology and various consequent metabolic characteristics during this period have been conducted in both vertebrate and invertebrate animals. The sea cucumber Apostichopus japonicus especially the large and mature individuals aestivate when water temperatures rise above 20°C. Previous studies have shown that aestivation is temperature-dependent and may vary depending on location and body size. Limited information is available on aestivation physiology in Apostichopus japonicus. In the present study, the authors assayed the gut microbiota of the sea cucumber Apostichopus japonicus from different groups including aestivating adult, non-aestivating adult and young animals. The aestivating adult group showed close association with some genera which were proposed as biomarkers of this group. The authors also revealed the most important metabolic pathways during the aestivation period providing new insights into potential biological associations among gut microbiota, metabolites, and aestivation in Apostichopus japonicus. The results of this study are illustrated with relevant figures and tables. Statistical seems to be adequate and correctly used but some issues still exist. The discussion gives a comprehensive interpretation of the main findings. The paper is of interest to many ecologists dealing with echinoderm aquaculture and physiology of sea cucumbers. I have only minor suggestions to improve the ms.
Recommendations.
L 75-80. It was found… This section should be removed as it corresponds to “Conclusion”, not to “Introduction”.
Figure 1. Consider replacing “Last weight” with “Final weight”
Figures 2–6. Most of these figures have very small font sizes and it is impossible to understand what exactly is written.
Supplementary materials are cited but not uploaded or not available for reviewing.
Specific remarks.
L 15. Consider replacing “were significantly increased” with “significantly increased”
L 50. Consider replacing “research studies” with “studies”
L 423. Consider replacing “positive correlation, blue lines indicate negative” with “a positive correlation, blue lines indicate a negative”
